# Lifestyle Behaviours Profile of Spanish Adolescents Who Actively Commute to School

**DOI:** 10.3390/children10010095

**Published:** 2023-01-03

**Authors:** Evelyn Martín-Moraleda, Iván Pinilla-Quintana, Cristina Romero-Blanco, Antonio Hernández-Martínez, Fabio Jiménez-Zazo, Alberto Dorado-Suárez, Virginia García-Coll, Esther Cabanillas-Cruz, Maria Teresa Martínez-Romero, Manuel Herrador-Colmenero, Ana Queralt, Nuria Castro-Lemus, Susana Aznar

**Affiliations:** 1PAFS Research Group, Faculty of Sports Sciences, University of Castilla-La Mancha, 45004 Toledo, Spain; 2PAFS Research Group, Faculty of Nursing, University of Castilla-La Mancha, 13071 Ciudad Real, Spain; 3Postdoctoral fellow “Margarita Salas”, University of Murcia, 30001 Murcia, Spain; 4PROFITH “PROmoting FITness and Health through Physical Activity” Research Group, Sport and Health University Research Institute (iMUDS), Department of Physical Education and Sports, Faculty of Sport Sciences, University of Granada, 18011 Granada, Spain; 5“La Inmaculada” Teacher Training Centre, University of Granada, 18013 Granada, Spain; 6Department of Nursing, University of Valencia, 46010 Valencia, Spain; 7FENIX Research Group, Faculty of Sports Sciences, University of Sevilla, 41004 Sevilla, Spain; 8CIBER of Frailty and Healthy Aging (CIBERFES), 28029 Madrid, Spain

**Keywords:** active commuting, adolescents, lifestyle behaviours, diet, sleep, screen

## Abstract

The aim of this study was to study different ‘healthy profiles’ through the impact of multiple lifestyle behaviours (sleep patterns, screen time and quality diet) on active commuting to school (ACS) in adolescents. Sixteen secondary schools from four Spanish cities were randomly selected. All participants filled in an “Ad-Hoc” questionnaire to measure their mode of commuting and distance from home to school and their lifestyle behaviours. A multivariate logistic regression model was performed to analyse the main predictor variables of ACS. The final sample was 301 adolescents (50.2% girls; mean age ± SD: 14.9 ± 0.48 years). The percentage of ACS was 64.5%. Multiple logistic regressions showed: boys were more active commuters than girls [OR = 2.28 (CI 95%: 1.12–4.64); *p* = 0.02]; adolescents who lived farther had lower probability to ACS [OR = 0.74 (CI 95%: 0.69–0.80); *p* < 0.001]; adolescents who met sleep duration recommendations were more likely to ACS [OR = 3.05 (CI 95%: 1.07–8.69); *p* = 0.04], while with each hour of sleep, the odds of ACS was reduced [OR = 0.51 (CI 95%: 0.30–0.89); *p* = 0.02]; higher odds were shown to ACS in adolescents who have more adherence to MD [OR = 1.16(CI 95%: 1.00–1.33); *p* = 0.05]; and habitual breakfast consumption was inversely associated with ACS [OR = 0.41 (CI: 95%: 0.18–0.96); *p* = 0.04]. ACS was associated with being a boy, living at a shorter distance to school, a daily sleep time ≥ 8 h and presented a higher adherence to MD.

## 1. Introduction

The prevalence of obesity in adolescents has increased worldwide [1]. Some reports estimate 90% of adolescents with obesity will continue to suffer from obesity into adulthood [2]. For this reason, it is important for adolescents to acquire healthy lifestyle behaviours for the protection and promotion of their health, and to maintain these behaviours into their adulthood [3].

Despite its concerning upward trajectory, adolescent obesity is amenable to lifestyle modifications such as positive diet changes, physical activity (PA) and sleep to reduce the risk of developing chronic diseases in later life [4]. Therefore, it has shown the effects of adopting multiple healthy lifestyle behaviours, which may be greater than the sum of their individual impacts [5]. In fact, a more integrated focus recognises that 24 h movement behaviours are co-dependent. Also, a recent review suggests a relationship between adherence to the Mediterranean diet (MD) and some of these 24 h movement behaviours [6]. High PA, less sedentary behaviour, great adherence to a healthy diet and adequate sleep hours maintain a healthy body weight and reduce adolescent adiposity [6,7,8].

In Spain, 69.9% of adolescents did not meet the World Health Organization (WHO) recommendation for daily PA [9]. For this reason, active commuting to and/or from school (ACS) (e.g., on foot or by bicycle) is presented as an opportunity to increment PA levels and reduce obesity in adolescents [10]. Therefore, it could be interesting to analyse the relationship between ACS and other lifestyle behaviours (sedentary behaviours, sleep quality and balanced nutrition) to explore if there is a ‘healthy profile’ related to ACS. These lifestyle behaviours had been related individually with general PA or physical education (PE) [11,12]; however, there is paucity of research with all behaviours together in relation with ACS.

In particular, sedentary behaviours, in a study carried out by Aparicio-Ugarriza et al. (2020) [10] showed a negative and significant association between ACS and screen time in adolescents. Moreover, a systematic review on children and adolescents found that active commuters spent significantly less time sitting than sedentary commuters [13]. On the other hand, Tassitano et al. (2020) [14] showed that higher screen time, with a large proportion of time spent playing videogames, was related to ACS in adolescents. Sleep duration is considered a relevant healthy behaviour in adolescents because of its links with numerous chronic diseases [15,16]. Adolescents with inadequate sleep duration on school days reported a lower percentage of ACS [17,18,19]. Finally, scarce research has shown results in quality diet and its association with ACS in adolescents. Only Martinez-Gomez et al. (2011) [17] reported that Spanish adolescents with habitual breakfast consumption were less likely to commute actively to school. Other studies did not find any association between breakfast consumption [18] or fruit consumption [17] and ACS. 

Knowing the association of these behaviours all together with ACS could help to promote it to increase total PA. The null hypothesis of the study is that there is no relationship between lifestyle behaviours (i.e., screen time, sleep patterns and quality diet) and ACS. This evidence suggests the need to analyse more in depth the relationship between these behaviours with ACS. For this reason, the aim of this study was to study different ‘healthy profiles’ through the impact of multiple lifestyle behaviours on ACS in Spanish adolescents. 

## 2. Materials and Methods

### 2.1. Study Design

A cross-sectional study was conducted to investigate adherence to ACS in relation to environment, psychosocial variables and healthy lifestyle factors in Spanish adolescents. All measurements were carried out in 2020 as part of the “PACO & PACA” National Research Project “Cycle and Walk to School, Cycle and Walk to Home” (ref: PGC2018-099512-B-I00). The study design, protocols and methodology of the PACO & PACA Project were approved by the Ethics Committee for Research with Drugs-SESCAM (ID: C-392) and in accordance with the Helsinki Declaration (1961) revised in Fortaleza (2013).

### 2.2. Participants and Recruitment

Sixteen secondary schools from four Spanish cities (Toledo, Granada, Sevilla and Valencia) were randomly selected. To ensure sample representativeness, socio-economic status (SES) and walkability levels of the neighbourhood were considered in the sampling processes. SES was calculated with population educational level obtained from National Statistics Institute and schools were classified as low SES or high SES. After, using a geographical information system, walkability was calculated for each school included in the study and they were classified as low walkability or high walkability. From this classification, in each city, one centre from each category (low SES-low walkability; low SES-high walkability; high SES-high walkability; high SES-low walkability) was selected through random probabilistic sampling. 

For adolescents to be included in the study, the inclusion criteria were: (a) being in the 3rd grade of secondary school, (b) returning the signed consent form before the start of the study, (c) being present on the day of data collection (d) having complete data of the variables used in this study and (e) living at a distance less than 4500 m from the school [20]. The initial sample consisted of 549 adolescents in the 3rd level of secondary school. Eighty-five adolescents were excluded based on the inclusion criteria (a–d) and 165 were excluded after analysis due to their home being situated farther than 4500 m from the school (e) (Figure 1). 

### 2.3. Instruments

For this study, all participants filled in an “Ad-Hoc” questionnaire (PACO&PACA Questionnaire) to measure their mode of commuting and distance from home to school and their lifestyle behaviours. All the scales included in the questionnaire were psychometrically studied previously.

### 2.4. Measures

#### 2.4.1. Mode of Commuting and Distance to School

The PACO&PACA Questionnaire includes the *Mode and Frequency of Commuting To and From School Questionnaire* from the PACO Questionnaire, which has been previously validated with an accelerometer in Spanish children and adolescents [21] and its reliability has been assessed [22]. 

Additionally, the questionnaire included general, social and demographic data, such as the home address. Distance between home and school was calculated objectively through Google Maps^TM^, selecting the shortest on-foot network path between each adolescent’s home address and the school, measured in meters [23]. 

#### 2.4.2. Lifestyle Behaviours

Adolescents provided data on their screen time spent in four different devices during weekdays: mobile phone, TV, computer or console games. It was evaluated using the Healthy Lifestyle in Europe by Nutrition in Adolescence (HELENA) (study’s screen time-based sedentary behaviour questionnaire) [24]. Each adolescent selected one of the following categories/day: (a) none, (b) less than half an hour, (c) between half an hour to an hour, (d) between one and two hours, (e) between two and three hours, (f) more than three hours. To unify screen time, we assigned a score to each possible answer (i.e., adolescents who spent 0 h scored 1 point, those who spent up to 30 min scored 2 points, between 30 min up to 1 h scored 3 points, between 1 h to 2 h scored 4 points, between 2 h to 3 h scored 5 points and finally, and adolescents who spent more than 3 h scored 6 points). Therefore, the minimum score started in 4 points (i.e., no time spent in any device), and the maximum score was 24 points. A new variable was created for this purpose, and it was named ‘screen variable composite’ including the sum of the punctuations from the four behaviours measured. 

Behaviours related to sleep were gathered by questions about the time when they went to sleep, and the time when they woke up from the PASOS (Physical Activity, Sedentarism and Obesity in Spanish Youth) Study Questionnaire [9]. They self-reported the exact time when they went to sleep and when they got up on a usual school day, and sleep duration was calculated on this basis. Adolescents were categorised as having “adequate sleep duration” when they met sleep time recommended by the Sleep Foundation, and “non-adequate sleep duration” when they did not meet the recommendations. According to the recommendations for young and old adolescents it was from 8 to 10 h [25]. 

The KIDMED (Mediterranean Diet Quality Index for children and adolescents) questionnaire was used to evaluate adherence to the MD in adolescents [26]. It consists of 16 items, where there are 4 questions denoting a negative connotation to the MD (i.e., consumption of fast food, baked goods, sweets and skipping breakfast) and 12 questions denoting a positive connotation (i.e., consumption of oil, fish, fruits, vegetables, cereals, nuts, pulses, pasta or rice, dairy products and yoghurt). Questions denoting negative connotation are scored with −1, while positive connotation questions are scored with +1. According to the KIDMED Index, a score of 0–3 reflects poor adherence to the MD, a score of 4–7 describes average adherence, and a score of 8–12 good adherence [26,27]. Daily breakfast was asked with the following question from the PACO Questionnaire [28]: “From Monday to Friday during the school year, how many days do you have breakfast?”. The participants could report: “5 days”, “4 days”, “3 days”, “2 days”, “1 day”, or “I never have breakfast on school days”. Students were categorised as “skipping breakfast” when adolescents reported eating breakfast on fewer than 5 days and “daily breakfast” if they reported having breakfast on all 5 days in a usual school week.

### 2.5. Procedure

Adolescents and their parents/guardians and physical education teachers were fully informed verbally and in writing about the nature and purpose of the study. All parents/guardians signed an informed consent form prior to participation.

Adolescents were tested in a session during class time following the COVID-19 secondary education protocol in each Spanish region. An experienced researcher first introduced the online questionnaire to students, explained the procedure for completing the survey and personally answered all participants’ questions. 

### 2.6. Statistics Analysis

Descriptive characteristics were presented as adjusted means with standard deviation (SD) for quantitative variables or as a proportion for categorical variables. Quantitative data was checked for normality using skewness and kurtosis normality tests. 

Chi-square statistical analysis was performed to evaluate differences between mode of commuting to school and related factors. When independent variables were ordinal (screen time) a Mann–Whitney U Test was also analysed. 

A multivariate logistic regression model was performed to analyse the main predictor variables of ACS, including age, gender, screen time, sleep duration, breakfast consumption habits, KIDMED punctuation and distance from home to school. A logistic regression model was fitted to determinate which variables are part of the predictive model. Firstly, a bivariate analysis was performed and secondly the multivariate analysis.

The predictive power of the model was analysed by plotting the corresponding Area under the Receiver Operating Characteristic (ROC) curve. The adjusted Odds Ratio (aOR) was estimated with a Confidence Interval (CI) of 95%. In order to assess the prediction in qualitative terms, the Swets’s criteria, whose values range from 0.5–0.6 (bad), 0.6–0.7 (poor), 0.7–0.8 (satisfactory), 0.8–0.9 (good), and 0.9–1.0 (excellent) were used. The analysis was conducted using SPSS, IBM v. 25.0 for Windows. The level of significance was set at *p* ≤ 0.05. 

## 3. Results

The final sample consisted of 301 adolescents (50.2% girls; mean age ± SD: 14.9 ± 0.48 years) meeting the inclusion criteria. The distribution of the sample among the four cities is presented in Table 1. Moreover, the mean distance between home and school for adolescents is 1275 ± 901.42 m (Table 1). A total of 64.5% of the adolescents commuted actively to and from school and their mode of commuting was mainly ‘walking’ (63.5%). Among the passive modes of commuting, the use of the car is the most common (around 30%).

Regarding screen time and mode of commuting to school, there were no significant differences at any screen (Table 2). Mobile phone was the most used screen, being used more than 3 h per day by 36% of the participants, independently of their mode of commuting.

Table 3 presents associations between ACS and behavioural factors (i.e., screen time, sleep patterns and nutritional habits). The multiple logistic regressions showed that those adolescents who are boys are twice more likely to commute actively to school than girls [Odds Ratio (OR) = 2.28 (CI 95%: 1.12–4.64); *p* = 0.02], and those adolescents who live farther were less likely to commute actively to school [OR = 0.74 (CI 95%: 0.69–0.80); *p* < 0.001]. Adolescents who met the sleep duration recommendations (i.e., sleep at least 8 h) were three times more likely to commute actively to school than those who did not [OR = 3.05 (CI 95%: 1.07–8.69); *p* = 0.04], while for each hour of sleep, the odds of ACS was reduced [OR = 0.51 (CI 95%: 0.30–0.89); *p* = 0.02]. Another related factor was adherence to the MD (KIDMED test punctuation), showing higher odds to ACS in adolescents who had more adherence to the MD [OR = 1.16(CI 95%: 1.00–1.33); *p* = 0.05]. These analyses also showed that habitual breakfast consumption was also inversely associated with ACS [OR = 0.41 (CI: 95%: 0.18–0.96); *p* = 0.04]. There were no associations between ACS and the screen variable composite (*p* = 0.44). 

The model had an AUC_ROC_ (95% CI) of 0.90 (0.86–0.93), which is considered excellent according to Swets’s criteria and a Nagelkerke’s R2 value of 0.58. 

## 4. Discussion

The current study analysed the relationship between ACS and several lifestyle behaviours such as screen habits, sleep patterns and quality diet. Additionally, this research was the first that had aimed to examine the influence between diet quality and ACS in adolescents. The main findings suggest that ACS is associated with the following: boys, live a shorter distance to school, have a daily sleep ≥ 8 hours and present a higher adherence to MD. 

Several factors could also affect decisions in respect to the choice in the mode of commuting to school (individual, social, environmental and politics) [29]. In our study, we explored individual factors such as screen habits, sleep patterns and nutrition patterns. Additionally, distance between home and school was included due to it being one of the most important environmental factors related to ACS [30].

### 4.1. ACS and Gender

Gender is one of the factors that most affect PA in Spain [31]. In this study, boys were more active commuters than girls, as we can corroborate with others studies [32,33]. Therefore, in addition to sex-related differences in the prevalence of ACS, the amount of PA accumulated during ACS may be different in adolescent males versus females [34].

### 4.2. ACS and Distance between Home and School

Previous studies showed distance between home and school as the most related factors in the election of mode of commuting [32,35]. D’Haese et al. (2011) [36] established achievable distances to walk (1500 m) and bike (3000 m) to school and a Spanish article found the threshold distance for walking to school was 1350 m or 0.84 miles in adolescents [37]. Our study highlights an average distance lower than 4500 m to promote ACS and active commuters had their homes significantly closer to school in line with other studies [38].

### 4.3. ACS and Lifestyle Behaviours

Due to the lack of evidence that analysed lifestyle behaviours (sedentary behaviours, sleep patterns and quality diet) all together with ACS, in the following paragraphs we have discussed the evidence of the association between each behaviour with ACS independently.

#### 4.3.1. Sedentary Behaviours

This study did not find associations between ACS and sedentary behaviours during weekdays. Likewise, Martinez-Gomez et al. (2011) [17] showed that sedentary behaviours such as television viewing and reading as a hobby were not significantly associated with ACS in Spanish adolescents. Moreover, Aparicio-Ugarriza et al. (2020) [10] found sedentary behaviours were not related to ACS, but a negative and significant association was observed between ACS and time spent studying without internet use in boys. 

On the other hand, a recent study in different WHO regions [39] showed ACS was inversely and moderately associated with sedentary behaviours in over half of studied countries. In this study, ACS reduced the odds of adolescents reporting high sedentary behaviours by 34% in low-income countries, 16% to 19% in middle income countries and 14% in high-income countries. Therefore, a country’s income could have a role in this association.

#### 4.3.2. Sleep Patterns

The current study suggests that adolescents who meet sleep duration recommendations (between 8 and 10 h per day) presented more ACS. Previous studies [17,18] also indicated that adequate sleep duration was associated with ACS in adolescents. However, when we consider the total number of hours of sleep in our study, ACS was inversely associated with a longer sleep duration. This implies that exceeding the sleep duration recommendations may not be beneficial to promote ACS. There is controversy in these results because the data presented by another study [40] suggested ACS was associated with further hours of sleep. However, these results showed time in bed, and not necessarily sleep time. Moreover, the same study reported that longer trips in ACS were associated with a reduction of sleep duration. Interventions concerning ACS must be carried out cautiously to avoid causing a reduction in sleep time [40]. 

#### 4.3.3. Diet Quality

According to our multivariate logistic regression analysis, adolescents who had higher adherence to the MD showed higher probability to ACS. A study found associations between MD with daily PA and mode of commuting to other places in the city different to the school among adolescents [41]. The rate of adherence to the MD decreased with the increase of hours of inactivity and with the use of car or mopped as the most frequent mode of commuting [41]; however, no study exists that analyses the relation between ACS and adherence to MD.

Another relevant finding in our study showed that habitual breakfast was inversely associated with ACS. The same association has also been confirmed by other studies [17,18], where adolescents hypothetically might prefer to spend more time eating breakfast rather than commute actively to school or those who commute passively to school may have more time to have breakfast. Future research is needed in this regard. 

### 4.4. Limitations and Strengths

The primary limitation of this study is inherent to its cross-sectional nature and consequently, we cannot confirm whether the modes of commuting to school determine an adequate sleep duration, better adherence to MD and daily breakfast habit or vice versa. A future longitudinal study design is recommended. Another limitation is that the data was self-reported, which may result in an under or over estimation of sleep time and adherence to MD.

However, to the best of our knowledge, this is the first study analysing data about the commuting behaviours of Spanish adolescents and their relationship to sleep duration, adherence to MD and daily breakfast consumption (talking into account distance from home to school), which are important daily behaviours for the health of adolescent school students. This information can help us to understand and promote ACS. Although we performed a random sampling of schools, taking into account SES and neighbourhood walkability, it is a Spanish sample with its particular culture bias. Therefore, we cannot generalise these results.

## 5. Conclusions

In conclusion, in a sample of 301 adolescents belonging to 16 schools from 4 Spanish cities, ACS was associated with the following: being a boy, living at a shorter distance to school, having a daily sleep ≥ 8 h and having a higher adherence to MD. This study supports taking into consideration lifestyle behaviours when developing public health strategies that promote ACS among adolescents. 

## Figures and Tables

**Figure 1 children-10-00095-f001:**
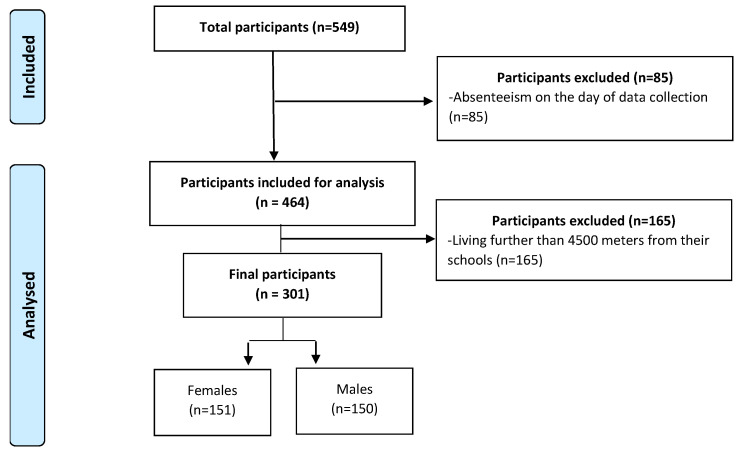
Study selection flow chart.

**Table 1 children-10-00095-t001:** Participant characteristics.

Variable	% (n)	Mean (SD)
**Age (years old)**		14.9 (0.48)
**City**		
Toledo	34.2 (103)	
Sevilla	17.6 (53)	
Granada	27.2 (82)	
Valencia	20.9 (63)	
**Girls**	50.2 (151)	
**Distance to school (meters)**		1275 (901.42)
**Media sleep hours (hours)**		7.9 (1.04)
**Meet sleep recommendations ≥ 8 h**	56 (17.2)	
**KIDMED (mean points)**		6.3 (2.65)
**Breakfast every day**	67.8 (204)	
**ACS**	64.5 (194)	
**Usual mode of commuting to and from school**		
Walk	63.5 (191)	
Car	29.6 (89)	
Scholar bus	3.0 (9)	
Public bus	2.3 (7)	
Bike	0.7 (2)	
Underground	0.7 (2)	
Scooter	0.3 (1)	

SD: Standard Deviation.

**Table 2 children-10-00095-t002:** Relationship between screen time and ACS.

Media Use	ACS	Screen Time Per Day						2	Mann Whitney–U
		None	Less than 30 min	Between 30–60 min	Between 1–2 h	Between 2–3 h	More than 3 h	*p* Value	*p* Value
TV viewing	Non-ACS	9.4 (10)	29.2 (31)	25.5 (27)	17.9 (19)	9.4 (10)	8.5 (9)	0.35	0.83
	ACS	16.0 (31)	23.7 (46)	18.6 (36)	18.0 (35)	12.9 (25)	10.8 (21)		
	Missing = 1								
Videogames	Non-ACS	43.0 (46)	20.6 (22)	15.0 (16)	10.3 (11)	5.6 (6)	5.6 (6)	0.34	0.96
	ACS	47.9 (92)	12.5 (24)	10.9 (21)	15.1 (29)	6.3 (12)	7.3 (14)		
	Missing = 2								
Mobile pone	Non-ACS	0.9 (1)	5.6 (6)	15.9 (17)	20.6 (22)	19.6 (21)	37.4 (40)	0.69	0.85
	ACS	1.0 (2)	7.8 (15)	9.8 (19)	21.8 (42)	23.3 (45)	36.3 (70)		
	Missing = 1								
Computer	Non-ACS	12.3 (13)	18.9 (20)	17.9 (19)	20.8 (22)	13.2 (14)	17.0 (18)	0.97	0.85
	ACS	14.5 (28)	17.1 (33)	16.6 (32)	18.7 (36)	13.5 (26)	19.7 (38)		
	Missing = 2								

**Table 3 children-10-00095-t003:** Bivariate and multivariate associations between ACS and sociodemographic features and lifestyle behaviours (n = 301).

	Mode of Transport		Bivariate Analysis	Multivariate Analysis	
Variable	Non Active n = 107	Active n = 194	OR 95% CI	aOR 95% CI	*p*-Value
**Age (years) Mean (SD)**	14.89 (0.50)	14.92 (0.47)	1.16 (0.70, 1.92)	1.58 (0.77, 3.24)	0.21
**Gender**					0.02
Girls	40.7 (61)	59.3 (89)	1 (Ref.)	1 (Ref.)	
Boys	30.5 (46)	69.5 (105)	1.56 (0.97, 2.52)	2.28 (1.12, 4.64) *	
**Distance from home to school (hectometres)**	20.21 (10.18)	8.63 (4.63)	0.77 (0.72, 0.82) **	0.74 (0.69, 0.80) **	<0.001
**Sleep hours Mean (SD)**	8.04 (0.97)	7.82 (1.07)	0.81 (0.64, 1.02)	0.51 (0.30, 0.89) *	0.02
**Sleep recommendations**					0.04
<8 h	36.3 (49)	63.7 (86)	1	1	
≥8 h	34.9 (58)	65.1 (108)	1.06 (0.66, 1.71)	3.05 (1.07, 8.69) *	
**KIDMED punctuation** **Mean (SD)**	6.21 (2.45)	6.33 (2.75))	1.02 (0.93, 1.11)	1.16 (1.00, 1.33) *	0.05
**Take breakfast every day**					0.04
No	29.9 (29)	70.1 (68)	1	1	
Yes	38.2 (78)	61.8 (194)	0.69 (0.41, 1.16)	0.41 (0.18, 0.96) *	
**Screen variable composite (sum of punctuations)** **Mean (SD)**	13.6 (3.83)	13.86 (4.21)	1.01 (0.96, 1.07)	0.97 (0.88, 1.06)	0.44

OR: Odds Ratio. aOR: Adjusted Odds Ratio by multivariate analysis * *p* ≤ 0.05, ** *p* < 0.001.

## Data Availability

Not applicable.

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
