# Peer review of "Lifestyle Behaviours Profile of Spanish Adolescents Who Actively Commute to School"

_children, 2023, doi:10.3390/children10010095_

Round 1
Reviewer 1 Report
Introduction
The authors provide a comprehensive rationale for the manuscript. However, there is no theoretical basis to explain young people's behaviours and especially the transfer of behaviours between similar contexts linked to physical activity (PA) and healthy behaviours.
Another substantial element is the novelty of the study. In this sense, there are similar studies that have analysed the transfer of PA to multiple contexts (e.g., PE <-> PA; PE -> Sport; etc.) and also the relationship between PA and PE with respect to healthy behaviours, for example between PA and healthy eating [1-3] or PE and healthy eating [4-7]. What is different about your study that others have not addressed? Only the relationship between PA behaviour (walking to school) and healthy behaviours?
PE = Physical Education
On the other hand, what are the hypotheses of the study? they should appear right after the main objective.
Discussion
This section should be completely redone. This section should be done in a single body. The contributions made by the authors are scarce (if not non-existent) and a mere extension of the results of the study. In this section, the results achieved in the study should be explained in a reasonable way, based on international literature (e.g. previous theories and studies).
1. Llargues, E., Franco, R., Recasens, A., Nadal, A., Vila, M., Pérez, M. J., ... & Castells, C. (2011). Assessment of a school-based intervention in eating habits and physical activity in school children: the AVall study. J Epidemiol Community Health, 65(10), 896-901.
2. Rodríguez, F., Palma, X., Romo, A., Escobar, D., Aragú, B., Espinoza, L., ... & Gálvez, J. (2013). Eating habits, physical activity and socioeconomic level in university students of Chile. Nutricion hospitalaria, 28(2), 447-455.
3. Turconi, G., Guarcello, M., Maccarini, L., Cignoli, F., Setti, S., Bazzano, R., & Roggi, C. (2008). Eating habits and behaviors, physical activity, nutritional and food safety knowledge and beliefs in an adolescent Italian population. Journal of the American College of Nutrition, 27(1), 31-43.
4. Popławska, H., Dmitruk, A., Kunicka, I., Dębowska, A., & Hołub, W. (2018). Nutritional habits and knowledge about food and nutrition among physical education students depending on their level of higher education and physical activity. Polish Journal of Sport and Tourism, 25(3), 35-41.
5. Trigueros, R., Mínguez, L. A., González-Bernal, J. J., Aguilar-Parra, J. M., Soto-Cámara, R., Álvarez, J. F., & Rocamora, P. (2020). Physical education classes as a precursor to the Mediterranean diet and the practice of physical activity. Nutrients, 12(1), 239.
6. Lirola, M. J., Trigueros, R., Aguilar-Parra, J. M., Mercader, I., Fernandez Campoy, J. M., & del Pilar Díaz-López, M. (2021). Physical education and the adoption of habits related to the Mediterranean diet. Nutrients, 13(2), 567.
7. González-Valero, G., Ubago-Jiménez, J. L., Ramírez-Granizo, I. A., & Puertas-Molero, P. (2019). Association between motivational climate, adherence to mediterranean diet, and levels of physical activity in physical education students. Behavioral sciences, 9(4), 37.
Reviewer 2 Report
I would like to thank you for giving me this opportunity to evaluate this scientific paper, which focuses on a topic of importance: the lifestyle behaviors (sleep patterns, screen time and quality diet) of adolescents who active commuting to school (ACS) in Spain.
This manuscript reports new findings and is theoretically based on the current literature.
The subject is very important for nutrition research in pregnancy.
- The manuscript is within the journal's scope.
- This study was well designed, executed, and presented.
- Figures and tables are well presented
- The conclusion is consistent with the evidence presented
- The discussion is relevant
- References are up to date and relevant
In the abstract pag 1 raw 35 the authors mentioned ‘’ had a diary sleep ≥8 hours ’’ it is daily sleep?
Also pag 1 R82’’ psico-social variables’’ it is psychosocial?
In the references nr 31 and 38 the name of Journal is not abbreviated.
Reviewer 3 Report
The paper is mainly descriptive and focused on its (not fully supported) conclusions, not adequately acknowledging the limitations of the study. The strengths and limitations of the study should be deeply addressed, taking into account sources of potential bias or imprecision: Discuss both direction and magnitude of any potential bias.
The following pertinent reports should be mentioned:
Sedentary patterns and cardiometabolic risk factors in Mexican children and adolescents: analysis of longitudinal data.
Int J Behav Nutr Phys Act. 2022 Dec 1;19(1):143. doi: 10.1186/s12966-022-01375-0. PMID: 36456985 Are health promoting lifestyles associated with pain intensity and menstrual distress among Iranian adolescent girls? BMC Pediatr. 2022 Oct 5;22(1):574. doi: 10.1186/s12887-022-03639-x. PMID: 36199045 Relationship between physical activity, screen-related sedentary behaviors and anxiety among adolescents in less developed areas of China. Medicine (Baltimore). 2022 Sep 30;101(39):e30848. doi: 10.1097/MD.0000000000030848. PMID: 36181048 Levels and Patterns of Physical Activity and Sedentary Behaviour of Primary School Learners in Lagos State, Nigeria. Int J Environ Res Public Health. 2022 Aug 29;19(17):10745. doi: 10.3390/ijerph191710745. PMID: 36078465 Improving physical activity behaviors, physical fitness, cardiometabolic and mental health in adolescents - ActTeens Program: A protocol for a randomized controlled trial. PLoS One. 2022 Aug 9;17(8):e0272629. doi: 10.1371/journal.pone.0272629. eCollection 2022. PMID: 35944003 Correlations between Physical Fitness and Body Composition among Boys Aged 14-18-Conclusions of a Case Study to Reverse the Worsening Secular Trend in Fitness among Urban Youth Due to Sedentary Lifestyles. Int J Environ Res Public Health. 2022 Jul 19;19(14):8765. doi: 10.3390/ijerph19148765. PMID: 35886622 Relations of life-style with lipids, blood pressure and insulin in adolescents and young adults. The Cardiovascular Risk in Young Finns Study. Atherosclerosis. 1994 Dec;111(2):237-46. doi: 10.1016/0021-9150(94)90098-1. PMID: 7718026
Round 2
Reviewer 1 Report
Good job